# Association between Late-Onset Ménière’s Disease and the Risk of Incident All-Cause Dementia

**DOI:** 10.3390/jpm12010019

**Published:** 2021-12-31

**Authors:** Il Hwan Lee, Hyunjae Yu, Seung-Su Ha, Gil Myeong Son, Ki Joon Park, Jae Jun Lee, Dong-Kyu Kim

**Affiliations:** 1Department of Otorhinolaryngology-Head and Neck Surgery, Chuncheon Sacred Heart Hospital, Hallym University College of Medicine, Chuncheon 24252, Korea; ilhwanloves@hanmail.net (I.H.L.); chskk2000@hallym.or.kr (S.-S.H.); ssadoong2@hanmail.net (G.M.S.); voix712@hallym.or.kr (K.J.P.); 2Institute of New Frontier Research, Division of Big Data and Artificial Intelligence, Chuncheon Sacred Heart Hospital, Hallym University College of Medicine, Chuncheon 24252, Korea; yunow@hallym.or.kr (H.Y.); iloveu59@hallym.or.kr (J.J.L.); 3Department of Anesthesiology and Pain Medicine, Chuncheon Sacred Heart Hospital, Hallym University College of Medicine, Chuncheon 24252, Korea

**Keywords:** Ménière’s disease, dementia, hearing, vestibular, cohort studies

## Abstract

Studies reported an association between impaired hearing and vestibular function with the risk of dementia. This study investigated the association between Ménière’s disease (MD) and the risk of dementia using a nationwide cohort sample of data obtained from the South Korea National Health Insurance Service. The MD group (*n* = 496) included patients aged over 55 years and diagnosed between 2003 and 2006. The comparison group was selected using propensity score matching (*n* = 1984). Cox proportional hazards regression models were used to calculate incidence and hazard ratios for dementia events. The incidence of dementia was 14.3 per 1000 person–years in the MD group. After adjustment for certain variables, the incidence of dementia was higher in the MD group than in the comparison group (adjusted hazard ratio (HR) = 1.57, 95% confidence interval = 1.17–2.12). Subgroup analysis showed a significantly increased adjusted HR for developing Alzheimer’s disease (1.69, 95% confidence interval = 1.20–2.37) and vascular dementia (1.99, 95% confidence interval = 1.10–3.57) in the MD group. Patients with dementia experienced a higher frequency of MD episodes than those without dementia. Our findings suggest that late-onset MD is associated with an increased incidence of all-cause dementia, and it might be used as a basis for an earlier diagnosis of dementia.

## 1. Introduction

Dementia is a widespread neurological disease in the elderly, and Alzheimer’s disease and vascular dementia are the two most common types of dementia. Patients with dementia have decreased cognition. Thus, they possess a high risk of imbalance, falls, and loss of spatial orientation causing wandering behaviors that can greatly affect quality of life and contribute to social and family burdens [1,2,3]. Previous studies have suggested that advanced age, genetic profile, cardiovascular disease, obesity, smoking, depression, and low educational attainment increase the risk for dementia [4,5].

Recently, the loss of hearing and vestibular function has emerged as an important risk factor for dementia. Several studies have reported that the risk of dementia increases as the severity of hearing loss increases, and that a decrease in the saccule function is associated with an increase in the risk of Alzheimer’s disease [6,7,8,9]. Ménière’s disease is an inner ear disease that can lead to hearing loss and disequilibrium. Classic symptoms include pressure in the ear, hearing loss, vertigo, and tinnitus [10]. Some patients with Ménière’s disease experience profound stressful conditions owing to these multiple symptoms [11]. Some studies have demonstrated a significant association between Ménière’s disease and hippocampal volume [12,13]. However, to date, the relationship between Ménière’s disease and dementia, including Alzheimer’s disease and vascular dementia, has not yet been clarified.

Therefore, this study aimed to investigate the association between Ménière’s disease and dementia. Further, we examined the potential risk of dementia in patients with late-onset Ménière’s disease using a nationally representative sample from the National Sample Cohort (NSC) data provided by the Korean National Health Insurance Service (KNHIS) in South Korea since the use of a representative nationwide population sample allowed us to investigate the entire medical service utilization history of more than 1 million South Koreans and provided an opportunity to analyze the association among specific diseases while adjusting for clinical and demographic factors.

## 2. Materials and Methods

### 2.1. Study Design and Ethics

This retrospective, nationwide propensity score-matched cohort study used data from the national health claims database collected by the KNHIS. The study protocol was approved by the Institutional Review Board of Hallym Medical University, Chuncheon Sacred Hospital (IRB No. 2021-08-006), and the study adhered to the tenets of the Declaration of Helsinki. The requirement for written informed consent was waived since the KNHIS dataset used in the study comprised de-identified secondary data.

### 2.2. Study Population

Among the 1,025,340 patients, those with Ménière’s disease were identified using the relevant International Classification of Diseases, 10th revision (ICD-10) code. The inclusion criteria were inpatient or outpatient care for an initial diagnosis of Ménière’s disease (H81) between January 2003 and December 2006 and age >55 years at cohort entry. To further improve the accuracy of Ménière’s disease definition, only patients who had been diagnosed with Ménière’s disease more than three times between 2003 and 2006 were included. Meanwhile, we excluded patients who (1) were diagnosed with dementia before the first diagnosis of Ménière’s disease, (2) died due to any cause between 2002 and 2006 or by an accident after 2006, and (3) underwent brain and heart surgery between 2002 and 2013. Additionally, we had a washout period of 1 year (2002) in this study. 

The comparison group (non-Ménière’s disease) comprised of randomly selected, propensity score-matched participants without Ménière’s disease from the remaining cohort registered in the database (four participants without Ménière’s disease for each patient with Ménière’s disease). To enhance the power of the comparison group, we excluded patients who had been diagnosed with any vertigo-related disease between 2002 and 2013.

### 2.3. Study Variables

In the present study, we divided the study population into three age groups (55–64, 65–74, and ≥75 years), three income groups (low: ≤30%, middle: 30.1–69.9%, and high: ≥70% of the median), and three residential areas (1st area: Seoul, the largest metropolitan region in South Korea; 2nd area: other metropolitan cities in South Korea; and 3rd area: small cities and rural areas). Data on the comorbidities, including hypertension (I10–I15), diabetes mellitus (E10–E14), stroke (I60–I63), chronic kidney disease (N18), and disorders of lipoprotein metabolism and other lipidaemia (E78), were also obtained using the relevant diagnostic codes. A schematic of the study design is shown in Figure 1.

### 2.4. Outcome Measures

The health claims data of all the participants were examined for the development of dementia (Alzheimer’s disease (F00, G30), vascular dementia (F01), and others (F02, F03)) until December 2013. In addition, we included only patients who were diagnosed with dementia by a neurologist. The outcome measures were event (dementia) or all-cause mortality. However, if patients had no events and were alive on 31 December 2013 (the final date of data collection), we censored this time point (Table 1).

### 2.5. Sensitivity Test

Data of the patients who were diagnosed with dementia (Alzheimer’s disease (F00, G30), vascular dementia (F01), and others (F02, F03)) and those aged over 65 years between 2012 and 2013 were collected from the health claims database. We excluded patients with a history of brain or heart surgery between 2002 and 2013. The comparison group included patients who had not been diagnosed with dementia (between 2002 and 2013) and who were aged >65 years between 2012 and 2013. We also excluded patients with a history of brain or heart surgery between 2002 and 2013. The comparison group (one patient without dementia for every patient with dementia) was selected using propensity score matching according to age, sex, residential area, household income, and comorbidities (Table 2).

### 2.6. Statistical Analysis

The risk of dementia was compared between the Ménière’s disease group and the comparison group using person–years at risk, defined as the duration between either the date of Ménière’s disease diagnosis or the enrollment date of the comparison group (1 January 2005) and the patient’s respective endpoint. Incidence rates per 1000 person–years for dementia were obtained by dividing the number of patients with incidents of specific diseases by person–years at risk. To determine whether Ménière’s disease increased the risk of the occurrence of specific diseases, we used Cox proportional hazard regression analyses to calculate the hazard ratio (HR) and 95% confidence intervals (CI), adjusted for the other independent variables. Levene’s test was performed to test the homogeneity, and Welch’s analysis of variance was used to investigate the difference in the frequency of Ménière’s disease. All the statistical analyses were performed using R (version 3.5.0; R Foundation for Statistical Computing, Vienna, Austria). *p* = 0.05 was considered significant.

## 3. Results

In total, 496 patients and 1984 participants were categorized to the Ménière’s disease group and comparison group (non-Ménière’s disease), respectively. There were no significant differences in sex, age, residential area, household income, and comorbidities between the two groups, indicating that each variable was appropriately matched (Figure 2). The characteristics of the study population by group are shown in Table 3. The overall incidence of dementia was 14.3 per 1000 person–years in the Ménière’s disease group and 11.1/1000 person–years in the comparison group (Table 4). 

After adjusting for sex, age, residence, income level, and comorbidities, late-onset Ménière’s disease was significantly associated with the subsequent development of events related to dementia (adjusted HR = 1.57, 95% CI = 1.17–2.12). Additionally, the adjusted HR for developing Alzheimer’s disease (1.69, 95% CI = 1.20–2.37) and vascular dementia (1.99, 95% CI = 1.10-3.57) was significantly higher in the Ménière’s disease group than in the comparison group (Table 5). 

The Kaplan-Meier survival curves with log-rank tests for the 11-year follow-up period are presented in Figure 3. The risk of dementia was significantly higher in the Ménière’s disease group than in the comparison group (Figure 3A). 

Figure 3B and 3C presents the Kaplan–Meier survival curves with log-rank tests for the cumulative hazard plot of specific disease-free (Alzheimer’s disease and vascular dementia) between the comparison and Ménière’s disease groups. The results indicated that the patients with Ménière’s disease developed all-cause dementia, Alzheimer’s disease, and vascular dementia more frequently than those in the comparison group during the 11-year follow-up period. 

To test the sensitivity of our findings, we investigated the difference in the frequency of Ménière’s disease between the dementia and non-dementia patients. For this comparison, we selected the dementia group and the matched comparison group (Table 6), and then we retrospectively compared the frequency of Ménière’s disease events between the two groups (Table 7). The results showed a significantly higher frequency of Ménière’s disease events in the dementia group than in the comparison groups (Table 4).

## 4. Discussion

Dementia is a disease characterized by the profound loss of cognitive functioning that interferes with daily life and activities. Essentially, the incidence of dementia is projected to increase, thus making it a major public health problem. To date, several studies have reported that loss of hearing and vestibular function might be implicated in dementia [14,15,16,17]. However, the relationship between Ménière’s disease, an idiopathic chronic endolymphatic hydrops, and dementia, including Alzheimer’s disease and vascular dementia, remains unclear to date. In the present study, we found that the cumulative incidence of all-cause dementia was higher in patients with late-onset Ménière’s disease than in those without late-onset Ménière’s disease. Additionally, the HR of dementia, after adjusting for sex, age, residence, income level, and comorbidities, in patients with late-onset Ménière’s disease was significantly higher in patients with late-onset Ménière’s disease than in those without late-onset Ménière’s disease. Moreover, in the subgroup analysis, we observed that the adjusted HR of Alzheimer’s disease and vascular dementia was higher in patients with late-onset Ménière’s disease compared to the comparison group, respectively. These findings are consistent with those of previous studies showing a relationship between vestibular dysfunction and cognitive impairment, such as mild cognitive impairment and Alzheimer’s disease [9,18]. Other prior studies also described the association between hearing loss and dementia and presented a possible mechanism of hearing loss, such as increasing cognitive load, accelerating brain atrophy, and social disengagement [6,19,20,21].

It is well-known that the hippocampus is responsible for memory, learning, and emotion. As such, studies on neurodegenerative diseases have focused on the hippocampus. One study demonstrated that compared with controls, patients with bilateral vestibular loss had a significantly atrophic hippocampus (16% decreased relative to controls) [22]. Other studies also reported that the hippocampal volume of patients with Ménière’s disease was significantly smaller than that of controls [12,13]. Furthermore, chronic dizziness and accompanying ear symptoms in Ménière’s disease result in limitations in daily activities, resulting in emotional stress. The resulting stress could activate the hypothalamus-pituitary-adrenal axis, increasing the cortisol secretion, which in turn could affect the hippocampus and cause atrophic changes [23,24]. Increasing emotional stress is also associated with smaller hippocampal volume in patients with psychotic disorders [25]. It indicates that late-onset Ménière’s disease may be related to the subsequent development of events related to dementia.

We also confirmed the sensitivity of our results by comparing the difference in the frequency of Ménière’s disease between the dementia and non-dementia groups. This sensitivity test demonstrated that patients with dementia experienced an episode of Ménière’s disease more frequently before the dementia diagnosis. Interestingly, our data revealed that patients with late-onset Ménière’s disease had a slightly higher adjusted HR in vascular dementia than in Alzheimer’s disease. Several studies have shown that vascular events and oxidative stress affect inner ear homeostasis and contribute to the development of Ménière’s disease [26,27,28]. Additionally. vascular problems and oxidative stress are important risk factors for the development of vascular dementia [29,30] Thus, we hypothesized that these causal similarities between the two diseases may have caused the higher adjusted HR in vascular dementia than in Alzheimer’s disease.

Our study had several strengths. First, to the best of our knowledge, this is the first cohort study to use nationwide population-based data to evaluate all-cause dementia in patients with Ménière’s disease. Second, we had a relatively long observation period of 11 years. Third, to improve diagnostic accuracy, we only enrolled patients with dementia diagnosed by neurologists. Finally, the KNHIS-NSC data has fair to good reliability. A prior validation study of KNHIS-NSC data reported a similar prevalence of 20 major diseases in each year [31]. However, this study also has certain limitations. First, specific personal medical data, including body mass index, pathological findings, laboratory data, or behavioral risk factors (smoking or alcohol consumption history), could not be accessed. Second, patients with Ménière’s disease and dementia were identified based on ICD-10 diagnostic codes, not on detailed medical records that include information on medical history and neurocognitive examination results. Therefore, this study could have a misclassification bias. To overcome this issue, we exclusively included patients with Ménière’s disease diagnosed by otorhinolaryngologists and patients with dementia diagnosed by neurologists. Third, we did not investigate the effect of Ménière’s disease medication on the development of dementia. So, we could not access whether patients with well-controlled Ménière’s disease by medication such as isosorbide may be a lower incidence of subsequent dementia development or not. This analysis should be needed in future studies to enhance the medical practice. Fourth, the data regarding disease severity in Ménière’s disease, such as audiometric data, were lacking in our registry. Finally, because this was a retrospective cohort study, it was not possible to directly investigate and analyze the pathological mechanisms of Ménière’s disease and dementia. Future clinical studies that include a wider range of factors are needed to elucidate the underlying pathophysiological mechanisms.

## 5. Conclusions

The present study investigated a possible association between mid- and late-life patients with Ménière’s disease and the development of dementia. Late-onset Ménière’s disease is associated with an increased incidence of all-cause dementia, including Alzheimer’s disease and vascular dementia, providing new insights into the association between Ménière’s disease and dementia. Therefore, the results indicate that late-onset Ménière’s disease might be used as a basis for an earlier diagnosis of dementia. However, to evaluate the underlying exact pathophysiological mechanisms, further studies included a wider range of factors and diagnostic criteria are required.

## Figures and Tables

**Figure 1 jpm-12-00019-f001:**
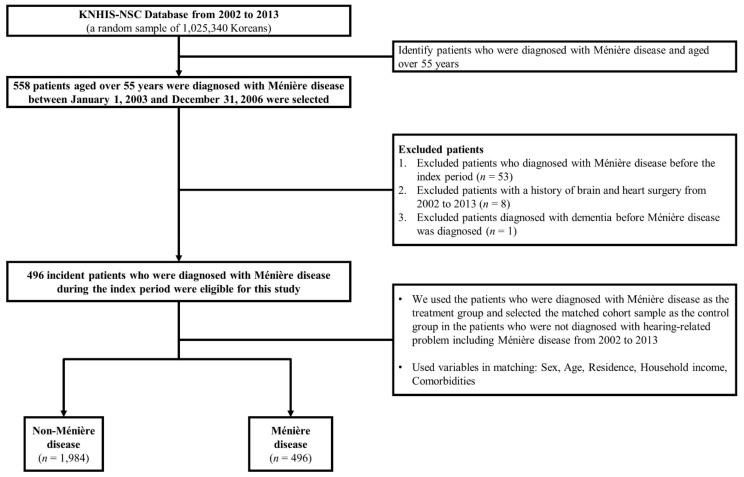
Schematic description of the study design.

**Figure 2 jpm-12-00019-f002:**
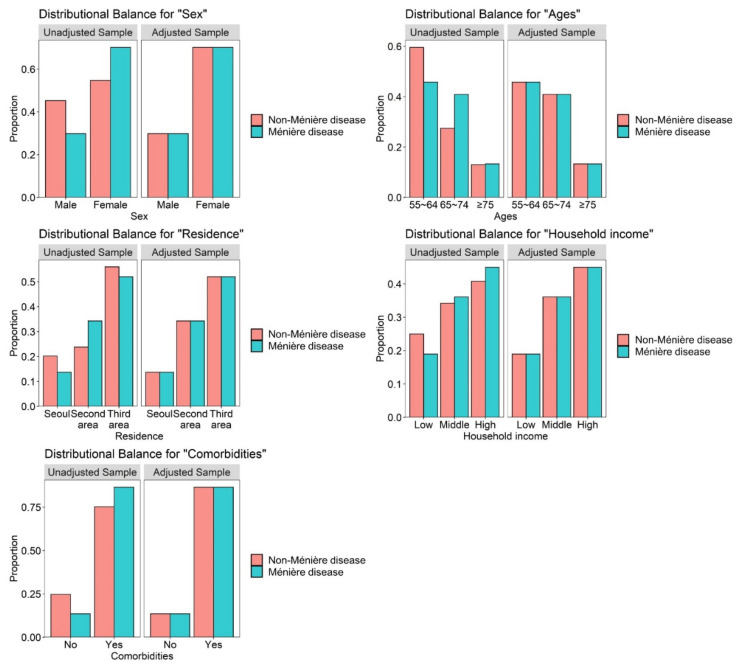
Balance plot for five variables before and after matching.

**Figure 3 jpm-12-00019-f003:**
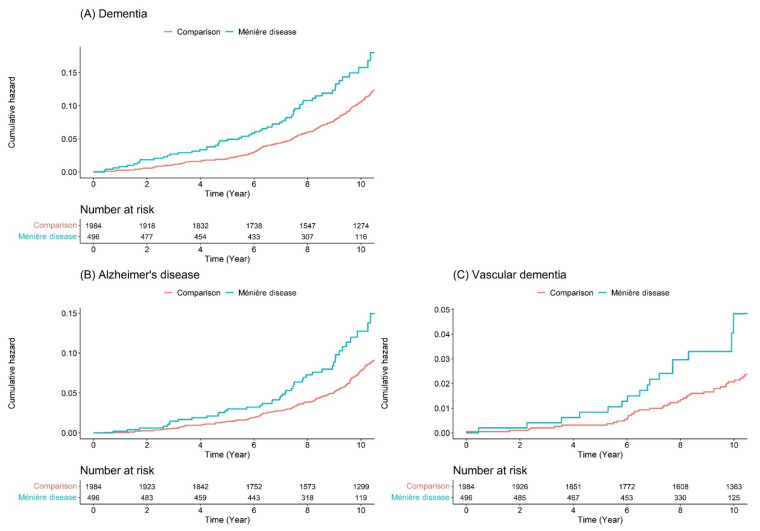
Cumulative hazard plot of dementia between the Ménière’s disease group and the comparison group. (**A**) All cause of dementia; (**B**) Alzheimer’s disease; (**C**) vascular dementia.

**Table 1 jpm-12-00019-t001:** Description of time to event and censored data.

	The Number of Dementia Event
**Event**	270
Comparison	212
Ménière’s disease	58
**Total censored (No event)**	2210
Comparison	1772
Ménière’s disease	438
**Termination of study**	1830
Comparison	1440
Ménière’s disease	390
**Loss to follow up/Drop-out**	380
Comparison	332
Ménière’s disease	48

**Table 2 jpm-12-00019-t002:** Detailed characteristics of the cohort dataset for sensitivity test.

Variables	Comparison (*n* = 18,706)	Dementia (*n* = 9353)	*p* Value
**Sex**			0.929
*Male*	5239 (28%)	2614 (27.9%)	
*Female*	13,467 (72%)	6739 (72.1%)	
**Ages (years)**			1.000
*65-74*	4260 (22.8%)	2130 (22.8%)	
*75-84*	9310 (49.8%)	4655 (49.8%)	
*≥85*	5136 (27.5%)	2568 (27.5%)	
**Residence**			0.588
*Seoul*	2562 (13.7%)	1281 (13.7%)	
*Second area*	4506 (24.1%)	2304 (24.6%)	
*Third area*	11,638 (62.2%)	5768 (61.7%)	
**Household income**			0.990
*Low (0–30%)*	6268 (33.5%)	3140 (33.6%)	
*Middle (30–70%)*	4140 (22.1%)	2064 (22.1%)	
*High (70–100%)*	8298 (44.4%)	4149 (44.4%)	
**Comorbidities**			0.696
*No*	2678 (14.3%)	1356 (14.5%)	
*Yes*	16,028 (85.7%)	7997 (85.5%)	

Seoul, the largest metropolitan area; second area, other metropolitan cities; third area, other areas.

**Table 3 jpm-12-00019-t003:** Characteristics of the study subjects.

Variables	Comparison (*n* = 1984)	Ménière’s Disease (*n* = 496)	*p* Value
**Sex**			1.000
*Male*	592 (29.8%)	148 (29.8%)	
*Female*	1392 (70.2%)	348 (70.2%)	
**Ages (years)**			1.000
*55-64*	908 (45.8%)	227 (45.8%)	
*65-74*	812 (40.9%)	203 (40.9%)	
*≥75*	264 (13.3%)	66 (13.3%)	
**Residence**			1.000
*Seoul*	272 (13.7%)	68 (13.7%)	
*Second area*	680 (34.3%)	170 (34.3%)	
*Third area*	1032 (52%)	258 (52%)	
**Household income**			1.000
*Low (0–30%)*	376 (19%)	94 (19%)	
*Middle (30–70%)*	716 (36.1%)	179 (36.1%)	
*High (70–100%)*	89–2 (45%)	223 (45%)	
**Comorbidities**			1.000
*No*	268 (13.5%)	67 (13.5%)	
*Yes*	1716 (86.5%)	429 (86.5%)	

Comparison, subjects without Ménière’s disease; Seoul, the largest metropolitan area; second area, other metropolitan cities; third area, other areas.

**Table 4 jpm-12-00019-t004:** Incidence per 1000 person–years and HR (95% CIs) of dementia between comparison (non-Ménière’s disease) and Ménière’s disease group.

Variables	*n*	Case	Incidence	Unadjusted HR (95% CI)	Adjusted HR (95% CI)
**Group**					
*Comparison*	1984	212	11.3	1.00 (ref)	1.00 (ref)
*Ménière’s* *disease*	496	58	14.3	1.55 (1.15–2.08) **	1.57 (1.17–2.12) **
**Sex**					
*Male*	740	60	9.1	1.00 (ref)	1.00 (ref)
*Female*	1740	210	13.0	1.40 (1.05–1.86) *	1.37 (1.03–1.83) *
**Ages (years)**					
*55* *−64*	1135	39	3.5	1.00 (ref)	1.00 (ref)
*65* *−74*	1015	154	16.3	4.68 (3.29–6.65) ***	4.58 (3.21–6.54) ***
*≥75*	330	77	34.8	12.16 (8.27–17.90) ***	12.35 (8.37–18.22) ***
**Residence**					
*Seoul*	340	31	9.8	1.00 (ref)	1.00 (ref)
*Second area*	850	101	13.0	1.34 (0.89–2.00)	1.20 (0.80–1.81)
*Third area*	1290	138	11.7	1.20 (0.81–1.77)	0.96 (0.65–1.42)
**Household income**					
*Low (0–30%)*	470	49	11.4	1.00 (ref)	1.00 (ref)
*Middle (30–70%)*	895	86	10.5	0.91 (0.64–1.29)	1.02 (0.72–1.45)
*High (70–100%)*	1115	135	13.1	1.13 (0.81–1.56)	1.09 (0.78–1.52)
**Comorbidities**					
*No*	335	27	9.0	1.00 (ref)	1.00 (ref)
*Yes*	2145	243	12.3	1.35 (0.91–2.01)	1.14 (0.76–1.70)

Seoul, the largest metropolitan area; second area, other metropolitan cities; third area, other areas. HR, hazard ratio; CI, confidence interval. * *p* < 0.05, ** *p* < 0.010, and *** *p* < 0.001.

**Table 5 jpm-12-00019-t005:** Incidence per 1000 person–years and HR (95% CI) of specific diseases (Alzheimer’s disease and vascular dementia).

Variables	*n*	Case	Incidence	Unadjusted HR (95% CI)	Adjusted HR (95% CI)
**Alzheimer’s disease**					
*Comparison*	1984	158	8.4	1.00 (ref)	1.00 (ref)
*Ménière’s* *disease*	496	44	10.6	1.67 (1.19–2.35) **	1.69 (1.20–2.37) **
**Vascular dementia**					
*Comparison*	1984	42	2.2	1.00 (ref)	1.00 (ref)
*Ménière’s* *disease*	496	16	3.8	2.02 (1.1–-3.63) *	1.99 (1.10–3.57) *

HR, hazard ratio; CI, confidence interval. * *p* < 0.05, ** *p* < 0.010.

**Table 6 jpm-12-00019-t006:** Detailed characteristics of the cohort dataset for sensitivity test.

Variables	Comparison (*n* = 18,706)	Dementia (*n* = 9353)	*p* value
**Sex**			0.929
*Male*	5239 (28%)	2614 (27.9%)	
*Female*	13,467 (72%)	6739 (72.1%)	
**Ages (years)**			1.000
*65-74*	4260 (22.8%)	2130 (22.8%)	
*75-84*	9310 (49.8%)	4655 (49.8%)	
*≥85*	5136 (27.5%)	2568 (27.5%)	
**Residence**			0.588
*Seoul*	2562 (13.7%)	1281 (13.7%)	
*Second area*	4506 (24.1%)	2304 (24.6%)	
*Third area*	11,638 (62.2%)	5768 (61.7%)	
**Household income**			0.990
*Low (0–30%)*	6268 (33.5%)	3140 (33.6%)	
*Middle (30–70%)*	4140 (22.1%)	2064 (22.1%)	
*High (70–100%)*	8298 (44.4%)	4149 (44.4%)	
**Comorbidities**			0.696
*No*	2678 (14.3%)	1356 (14.5%)	
*Yes*	16,028 (85.7%)	7997 (85.5%)	

Seoul, the largest metropolitan area; second area, other metropolitan cities; third area, other areas.

**Table 7 jpm-12-00019-t007:** Frequency table for Ménière’s disease episode between patients with dementia and non-dementia.

	Non-dementia (*n* = 18,706)	Dementia (*n* = 9353)
**Ménière’s** **disease**		
*0*	18,233 (97.5%)	9046 (96.7%)
*1*	252 (1.3%)	153 (1.6%)
*2*	74 (0.4%)	69 (0.7%)
*3*	47 (0.3%)	25 (0.3%)
*4*	24 (0.1%)	9 (0.1%)
*5*	19 (0.1%)	9 (0.1%)
*6*	14 (0.1%)	12 (0.1%)
*7*	8 (0%)	4 (0%)
*8*	6 (0%)	2 (0%)
*9*	7 (0%)	1 (0%)
*10*	2 (0%)	5 (0.1%)
*11*	2 (0%)	1 (0%)
*12*	0 (0%)	3 (0%)
*13*	4 (0%)	3 (0%)
*14*	3 (0%)	1 (0%)
*15*	2 (0%)	3 (0%)
*16*	1 (0%)	0 (0%)
*17*	1 (0%)	0 (0%)
*19*	0 (0%)	1 (0%)
*21*	0 (0%)	1 (0%)
*22*	1 (0%)	0 (0%)
*23*	1 (0%)	0 (0%)
*24*	0 (0%)	1 (0%)
*25*	0 (0%)	1 (0%)
*26*	1 (0%)	0 (0%)
*27*	1 (0%)	1 (0%)
*28*	1 (0%)	0 (0%)
*30*	0 (0%)	1 (0%)
*34*	0 (0%)	1 (0%)
*35*	1 (0%)	0 (0%)
*36*	1 (0%)	0 (0%)

## Data Availability

The authors confirm that the data supporting the findings of this study are available within the article.

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
