# Peer review of "Association between Late-Onset Ménière’s Disease and the Risk of Incident All-Cause Dementia"

_jpm, 2021, doi:10.3390/jpm12010019_

Round 1
Reviewer 1 Report
The authors described the association between late-onset Meniere's disease and the risk of incident all-cause dementia.
I think the results of this study are very interesting. However, the contents are insufficient for acceptance without revision and the manuscript needs to be revised in the following.
Materials and Methods
The patients with Meniere's disease (MD) were identified using the relevant International Classification of Diseases, 10th revision (ICD-10) code.
Is the diagnosis of MD according to the American Academy of Otolaryngology-Head and Neck Surgery (AAO-HNS) guide-lines and/or the Barany Society criteria?
Materials and Methods
The results with the MD group were compared with those with non-MD group and authors excluded patients who had been diagnosed with any hearing-related diseases.
It is said that dementia progresses in patients with age-related deafness by being cut off from the society. If authors compare those with MD, should authors select the patients with age-related deafness without MD as the control subjects?
Discussion
The authors did not investigate the effect of MD medication on the development of dementia.
As authors mentioned in the discussion, the difference between the presence and absence of treatment in not mentioned. Generally, patients with a definitive diagnosis of MD are treated by isosorbide. Can the authors add their thoughts in the part of discussion section?
Discussion
L 189
significantly higher in in patients with -----
Author Response
The authors described the association between late-onset Meniere's disease and the risk of incident all-cause dementia. I think the results of this study are very interesting. However, the contents are insufficient for acceptance without revision and the manuscript needs to be revised in the following.
Materials and Methods
The patients with Meniere's disease (MD) were identified using the relevant International Classification of Diseases, 10th revision (ICD-10) code. Is the diagnosis of MD according to the American Academy of Otolaryngology-Head and Neck Surgery (AAO-HNS) guide-lines and/or the Barany Society criteria?
Answer) Thank you for your comment. In South Korea, physicians usually used the criteria of definite Ménière’s disease to make a diagnosis. The diagnosis of definite Ménière’s disease developed by the collaboration of the Bárány Society, the Japan Society for Equilibrium Research, the European Academy of Otology and Neurotology (EAONO), the Equilibrium Committee of the American Academy of Otolaryngology-Head and Neck Surgery, and the Korean Balance Society is based on clinical criteria and requires the observation of an episodic vertigo syndrome associated with low- to medium-frequency sensorineural hearing loss and fluctuating aural symptoms (hearing, tinnitus and/or fullness) in the affected ear. Duration of vertigo episodes is limited to a period between 20 minutes and 12 hours. However, we could not acknowledge the exact proportion between definite and probable Ménière’s disease in diagnosis based on ICD-10.
Materials and Methods
The results with the MD group were compared with those with non-MD group and authors excluded patients who had been diagnosed with any hearing-related diseases. It is said that dementia progresses in patients with age-related deafness by being cut off from the society. If authors compare those with MD, should authors select the patients with age-related deafness without MD as the control subjects?
Answer) This is our mistake. Actually, we excluded patients who had been diagnosed with any vertigo-related diseases, not hearing-related diseases. We modified this typo error.
Discussion
The authors did not investigate the effect of MD medication on the development of dementia. As authors mentioned in the discussion, the difference between the presence and absence of treatment in not mentioned. Generally, patients with a definitive diagnosis of MD are treated by isosorbide. Can the authors add their thoughts in the part of discussion section?
Answer) We totally agreed with your opinion. Thus, we added more description regarding this issue in the section of Discussion as follows: “So, we could not access whether patients with well-controlled Ménière’s disease by medication such as isosorbide may be a lower incidence of subsequent dementia development or not. This analysis should be needed in future studies to enhance the medical practice.”
Discussion
L 189
significantly higher in in patients with -----
Answer) Thank you for your comment. As you commented, we modified this error.
Reviewer 2 Report
Dear authors,
thank you for submitting the paper with the topic: "Associations between late-onset Meniere`s disease and the risk of incident all-cause dementia".
The study was performed nicely, the methodes have been well selected and described. The interesting results have been extensively described and discussed and the references are extensive . The text is fluid to read. I have nothing to fault for a publication. Well done.
Author Response
Thank you for submitting the paper with the topic: "Associations between late-onset Meniere`s disease and the risk of incident all-cause dementia".
The study was performed nicely, the methodes have been well selected and described. The interesting results have been extensively described and discussed and the references are extensive. The text is fluid to read. I have nothing to fault for a publication. Well done.
Answer) Thank you for your kind comment.